# Evaluation of Selected Quality Parameters of “Agristigna” Monovarietal Extra Virgin Olive Oil and Its Apple Vinegar-Based Dressing during Storage

**DOI:** 10.3390/foods11081113

**Published:** 2022-04-13

**Authors:** Monica R. Loizzo, Vincenzo Sicari, Umile G. Spizzirri, Rosa Romeo, Rosa Tundis, Antonio Mincione, Fiore P. Nicoletta, Donatella Restuccia

**Affiliations:** 1Department of Pharmacy, Health and Nutritional Sciences, University of Calabria, 87036 Rende, Italy; monica_rosa.loizzo@unical.it (M.R.L.); g.spizzirri@unical.it (U.G.S.); rosa.tundis@unical.it (R.T.); fiore.nicoletta@unical.it (F.P.N.); donatella.restuccia@unical.it (D.R.); 2Department of Agraria, “Mediterranea” University of Reggio Calabria, Cittadella Universitaria, Località Feo di Vito, 89124 Reggio Calabria, Italy; rosa.romeo@unirc.it (R.R.); antonio.mincione@unirc.it (A.M.)

**Keywords:** extra virgin olive oil, Agristigna cultivar, oil-vinegar emulsion, CIELAB parameters, oxidative stability, sensorial analysis

## Abstract

This study aimed to investigate the quality parameters and the oxidative stability (180 days of storage) of a water–oil emulsion formulated with Calabrian (Italy) monovarietal Agristigna extra virgin olive oil and apple vinegar. The best extra virgin olive oil/apple vinegar *ratio* was found to be 85/15 (*v*/*v*) and lecithin (2% *w*/*v*) was the best additive to reach proper stability and viscosity over time. An increase of lightness parameters was evidenced in both products in a storage time-dependent manner. During storage, both oil and dressing showed a free acidity level beyond the accepted limit for extra virgin olive oil, whereas a slight increase of the peroxide value was observed only for Glasoil at the end of the observation time without affecting sensory attributes. A general decrease of phytochemicals was observed for extra virgin olive oil and Glasoil, with different reduction trends and degrees depending on the chemical class. A lower stability of Glasoil during shelf-life was confirmed by the worsening of the rheological features and by the polyunsaturated fatty acids reduction (up to −21.71%) with a corresponding increase of the monounsaturated fatty acids/polyunsaturated fatty acids ratio (about +25.69%).

## 1. Introduction

Extra-virgin olive oil (EVOO) is an essential condiment used by the population of the Mediterranean basin, with Greece, Italy and Spain as the most important producers and consumers worldwide [1]. EVOO phytochemical profile is affected by several factors including the geographical origin of fruits, as well as the technological, agronomical and environmental parameters [2]. In Italy, more than 538 *O. europea* cultivars are used to produce EVOO with a strong contribution of the Calabria region (southern Italy). Agristigna is one of the autochthons Calabrian cultivars resulting in a fruity oil, with a bitter taste and a spicy note of medium intensity [3]. Although olive oil is widely consumed as such, other products have been developed, including aromatized olive oils and dressings. In both products, EVOO is generally processed with spices, herbs or formulated with different ingredients (i.e., vinegar, essential oils, additives) to improve the starting oil nutritional value and/or its sensory characteristics, shelf-life and ease of use [4]. Among oil-based products, salad dressings represent ideal weight management options that have witnessed greater acceptance in recent years, mainly due to their organoleptic enhanced features and their aesthetic appeal. This trend is confirmed by the salad dressing market data: from their humble beginning, these products have now become mainstream, with a Compound Annual Grow Rate (CAGR) of 6.13% in the next few years (2020–2025) [5]. The development is mainly in terms of inclusion of organic and natural ingredients that assist in weight management, regulate blood pressure and a host of other health benefits. The salad dressings market has now evolved to a stage that versatile ingredients with clean-label and lifestyle claims are set to make greater gains to address the health and safety consciousness towards processed food products. However, the presence of ingredients other than EVOO should be carefully evaluated in terms of a product’s stability and maintenance of chemical and hygienic features over time. In this sense, the shelf-life of salad dressings should be at least 3 to 6 months at room temperature. Product failure modes include rancidity, presence of off-flavors, emulsion instability, discoloration, and microbial spoilage [6], although pourable dressing products are rarely associated with foodborne illness due to their acidic nature.

The scientific literature dealing with the shelf-life study of oil-based dressings appears somewhat limited.

The flow properties, stability rate and sensory characteristics of salad dressing emulsions prepared by modified canola lecithin (thermal treatment of lecithin-protein complex and ethanol treatment of lecithin protein complex), canola lecithin-canola protein isolate (CPI) complex and canola lecithin were investigated by Khalil et al. [7]. Dressing emulsions obtained by using a canola lecithin–CPI complex (treated by heat and ethanol), after two-month storage, was characterized by a better stability and sensorial characteristics than those prepared using unmodified canola lecithin. The typology and amount of vegetable oil involved, the emulsifiers used and the methods of mixing in the preparation of the emulsion seemed to be responsible of the dressing rheological behavior even if the oil used was soybean [8].

Successively, Perrechil et al. [9] investigated the emulsion stability of four different commercial Italian salad dressings formulated with vegetable oil and soybean oil and different other ingredients (i.e., oregano, onion, carrot, mustard, parsley, basil, celery, pepper, garlic, lemon juice, xanthan gum, vinegar, etc.). The analyses evidenced that the only stable salad dressing was that with the smallest droplet size. Moreover, it showed an overshoot at relatively low shear rates. This distinct rheological behavior was explained by the differences in composition, particularly related to the presence of a maltodextrin network. More recently, Abedinzadeh et al. [10] investigated the stability of a new formulation of salad dressing with olive oil and apple vinegar using different concentration of xanthan gum as emulsifier stored in the refrigerator for 3 months. In another work, Paraskevopoulou et al. [11] investigated the oxidative stability of “Greek salad dressing” formulated with olive oil, lemon juice and gum arabic or propylene glycol alginate in admixture with xanthan.

In this context, the main objective of this study was the evaluation of some quality and shelf-life features of an Agristigna EVOO-apple vinegar-based dressing over 180 days of storage at 4 °C. For this purpose, several determinations were carried out, including sensory, textural and oxidative analyses, CIELab measurements, total phenols, total flavonoids, total carotenoids, chlorophyl and vitamin E contents and fatty acid profile. Moreover, the emulsion physical characterization was also assessed by optical microscopy and dynamic light scattering measurements.

## 2. Materials and Methods

### 2.1. Chemicals and Reagents

Analytical-grade solvents were obtained from VWR International S.r.l., whereas chemicals and reagents were bought from Sigma-Aldrich S.p.a. (Milan, Italy).

### 2.2. EVOO and Preparation of W/O Emulsions

Agristigna EVOO was obtained using continuous mills by Frantoio “Meringolo” S.r.l. (Corigliano Calabro, Cosenza, Italy). EVOO samples have obtained the UNI10939, 2001 certification. “Glasoil” dressing was prepared by adapting different literature protocols with some modifications [7,10]. Lecithin (1–5% *w*/*v*) was dissolved in apple vinegar and mixed for one hour at 25 °C using a magnetic stirring bar. The w/o emulsions were prepared with an aqueous volume fraction ranging from 10 to 25% (*v*/*v*), which was dispersed into the Agristigna EVOO (90–75% *v*/*v*) by a silent crusher (Heidolph Silent Crusher M, Schwabach, Germany) at 18,000 rpm for 30 min. The resulting emulsion was poured into vials and stored in the dark conditions at 25 °C until further analyses. Both EVOO and Glasoil dressing were stored into green glass bottles and stored in the dark (at 10 °C) before analysis at 0, 45 and 180 days of storage.

### 2.3. Emulsion Physical Characterization

Emulsion droplets were observed at regular time intervals by an optical trinocular microscope (BA 300 POL, Motic Europe, Barcellona, Spain) equipped with a digital camera (Moticam 2, Motic Europe, Barcellona, Spain). Dynamic light scattering measurements were performed on emulsion samples placed in standard Q quartz glass cuvettes (optical path 10 mm, Mettler Toledo, Milan, Italy) by a Zetasizer Nano ZS instrument (Malvern Panalytical Ltd., Malvern, UK). The light intensity and its time autocorrelation function were measured at a 173° scattering angle. All measurements were performed at 25.0 ± 0.1 °C after 2 min of equilibration using automatic time settings. Polydispersity index (PDI) was obtained from the instrumental data fitting procedures using inverse Laplace transformation and Contin methods. PDI values lower than 0.3 indicated homogenous and mono-disperse populations.

### 2.4. Quality and CIELab Parameters

The free acidity value, the peroxide index and the fatty acid profiles of EVOOs were analyzed according to previously published methods [12]. CIELab parameters were measured using a PCE CSM-4 colorimeter (PCE, Lucca, Italy) [13]. Results were reported as lightness (L*), chroma (C*) and hue angle (H*).

### 2.5. Extraction of Bioactive Compounds and Determination of Total Phenols, Flavonoids, Carotenoids and Chlorophyll Content

Five g of EVOO or apple vinegar-based dressing (Glasoil) was extracted with 7:3 *v*/*v* of MeOH/H_2_O. The extract was treated with n-hexane and centrifuged for 10 min at 12,000× *g* rpm. The residue was taken up, filtered and stored at −20 °C until analysis [14]. Total phenols content (TPC) and flavonoids content (TFC) were determined as previously reported [13]. Both determinations were repeated in triplicate and expressed as mg/kg. For the total carotenoids content (TCC) analyses, samples were mixed with n-hexane (1:1, *v*/*v*) and the absorbance was read at 470 nm. Data are expressed as mean ± S.D. in mg/kg [15]. The total chlorophyll content was evaluated by applying the procedure described by Sicari et al. [16].

### 2.6. Determination of α-Tocopherol Content

To determine the of α-tocopherol concentration, samples were extracted following the method described by Bakre [17]. The extract was injected in Ultra-High-Performance Liquid Chromatography (UHPLC) PLATINblue (Knauer, Berlin, Germany) equipped with a binary pump system using a Knauer column C18 (1.8 μm, 150 × 3 mm) coupled with a Shimazu RF-20Axs Fluorescence Detector. The analytical column was kept at 30 °C and the fluorescence detector was set at 295 nm excitation wavelength and 325 nm emission wavelength. Methanol was employed as mobile phase, in an isocratic elution mode (flow: 1.0 mL/min; wavelength: 292 nm). The α-tocopherol was identified and quantified in comparison to external standard and the chromatograms elaborated through Clarity 6.2 software (Clarity System Limited, Totonto, ON, Canada).

### 2.7. Ultra-High-Performance Liquid Chromatography (UHPLC) Analysis of Agristigna EVOO

The extract obtained as previously described was injected in an UHPLC system (UHPLC PLATINblue), coupled with a PDA-1 (photodiode array detector; PLATINblue) for the determination of phenolic compounds and a fluorescence detector RF-20A/RF-20Axs model (Shimadzu Corporation, Kyoto, Japan) for the detection of α-tocopherols following the procedure indicated by De Bruno et al. [18]. Data were acquired with Clarity 6.2 software (Clarity System Limited, Totonto, ON, Canada) and expressed as mean ± S.D.

### 2.8. Fatty Acids Profile Determination

The fatty acids profile evolution during storage was determined by gas chromatography-mass spectrometry (GC-MS) analyses (Agilent, Milan, Italy). Before injection samples were derivatized as methyl esters (FAME) following the method described by Sicari et al. [16]. The FAMEs were identified by their retention times in comparison with an authentic standard mix containing FAMEs from C10 to C22.

### 2.9. Evaluation of EVOO and Its Apple Vinegar-Based Dressing Oxidative Stability

The oxidative stability of both Agristigna EVOO and its derived dressing (Glasoil) was investigated under accelerated condition by using OXITEST (VELP, Scientifica) [13]. Briefly, sample (5 g) was evenly distributed in hermetically sealed titanium chambers. The reactor temperature was setter at 90 °C and a pressure of 6 bar. The OXITEST response is the induction time (IT), which is automatically calculated by OXISoft™ software 3.0.0 (Velp Scientifica Srl – HQ, Usmate (MB), Italy).

### 2.10. Textural Analysis

Rheological tests were performed with a TA-XT Plus Texture Analyzer (Stable Micro Systems Ltd., Surrey, United Kingdom). Data acquisition and curves integration were carried out by Exponent software 6.1.4.0 (Stable Micro Systems Ltd., Surrey, United Kingdom). Samples were analyzed with a 50 mm diameter back extrusion container filled at 75%. Analysis was carried out immediately after the sample removal from refrigerator (5 °C storage temperature). Sample was compressed with a 45 mm diameter extrusion disc positioned centrally over the sample container. Operating parameters were test speed: 1 mm/s; post-test speed: 10 mm/s; distance: 15.00 mm; trigger force: 2.0 g; data acquisition rate: 200 pps. For each sample, five repetitions were evaluated. Testing was done at time 0, 45 and 180 days.

### 2.11. Sensory Analysis

A panel of eight assessors (males and females), from 25 to 60 years old, was recruited to assess the differences among the dressing samples at production day and after 45 and 180 days of storage. Following the method reported by Liang et al. [19] and Pau et al. [20], panelists were trained to distinguish and describe the aroma attributes. To recognize and identify the different aroma that occurred in apple vinegar-based dressing, the subjects were asked to perceive the aroma of standard solution (caffeic acid, glucose and sodium chloride citric acid) at different concentrations (from 0.8 to 4 g/L), ranking them according to taste perception intensity. Profile attributes were classified into three groups: appearance attributes (brightness and browning), aromatic aspects (typical aroma, sweetness and bitterness) and taste (typical taste, sweetness, bitterness and aftertaste). The evaluation was done using 10-point structured scales, where 0 is absent and 10 is extremely perceptible. The obtained data were used to define the sensory profile of each sample using the median values.

### 2.12. Statistical Analysis

Data are expressed as means ± standard deviation (S.D.) (*n* = 3). Statistical analyses were performed using SPSS software for Windows (SPSS Inc., Elgin, IL, USA) 22.0 Version. One-way analysis of variance test was used to evaluate the differences within and between groups followed by Tukey’s test to determine any significant difference in physical-chemical parameters (** *p* < 0.01) among investigated samples. Microsoft Excel 2010 software (Redmond, WA, USA) was used to calculate linear regression, assessment of repeatability, calculation of average, relative standard deviation, and *Pearson’s* correlation coefficient (*r*).

## 3. Results and Discussion

### 3.1. Emulsion Physical Characterization

From a physical-chemical point of view, emulsions are thermodynamically unstable systems that, after a certain period, separate into two immiscible phases. The increase in the shelf-life of an emulsion by kinetic stability is one of the main challenges of food formulation. To this regard, the addition of emulsifiers and stabilizers represents a valuable strategy to increase the kinetic stability of emulsions. Among emulsifiers commonly used in food industry, lecithin is one of the natural ingredients largely accepted by consumers, and generally recognized as safe by legislators [21]. Alternatively, polysaccharides, such as xanthan gum, gum arabic and tara gum were proposed as thickeners of emulsions to confer them long-term stability [22]. To achieve a stable w/o emulsion, different experimental parameters, such as the volume ratio between aqueous and organic phases, type and amounts of emulsifiers, were investigated in the present work. The tests clearly indicated that the best volume *ratio* between extra virgin olive oil and apple vinegar was equal to 85/15 (*v*/*v*), while lecithin (2% *w*/*v*) was the best emulsifier system. Different oil/vinegar *ratios* and different amounts of lecithin gave unstable emulsions. At the same time, the added lecithin amount was able to act as emulsion thickener, guarantying the required stability and conferring the appropriate viscosity for a long-term squeezable product. Alternatively, xanthan gum and tara gum (0.2–2% *w*/*v*) were tested, but no significant increase in term of emulsion formation and stability over time was recorded.

The quality of emulsion-based food products is deeply influenced by several parameters [23], and their stability generally refers to their aptitude to be unaffected or have slow changes in their serving properties over time. The foremost indicator for the stability of an emulsion is its average droplet size [24]. In this work, emulsion stability was monitored over time by optical microscopy and Dynamic Light Scattering Microscopy (DLS) measurements. In particular, the emulsion monitoring by optical microscopy highlighted a relatively good stability of the emulsions after 45 and 180 days, as is shown on the left side of Figure 1.

DLS measurements confirmed the good size homogeneity of samples and showed that the *w*/*o* emulsion consisted of a less abundant droplet population with an average diameter of around 530.2 ± 65.3 nm, accounting for a scattering intensity of 95.1 ± 3.7% and less than 1% for a size distribution in number, respectively. On the contrary, the predominant population had an almost 100% size distribution in number, as is evident from the right column of Figure 1. The most abundant droplet population showed an initial average diameter of around 26.8 ± 6.9 nm, accounting for a scattering intensity of 4.9 ± 0.7%. Such an average diameter increased after 45 days up to 42.3 ± 2.9 nm, reaching 423.4 ± 46.0 nm after 180 days, as a consequence of coalescence processes occurring over such a long-term period. In particular, the results indicated the coalescence of smaller into larger droplets, with an average diameter similar to that shown by the initial less abundant droplet population.

### 3.2. Evolution of Quality Parameters of Dressing during Storage

Free acidity, peroxide, CIELab parameters as well as oxidative stability were monitored for 180 days of storage at −4 °C for both Agristigna EVOO and derived Glasoil dressing. The highest quality EVOOs must feature a free acidity lower than 0.8%. In our case, the free acidity started from 0.31%, reaching 0.52% at the end of observational period, whereas peroxide values were in the range 12.14–17.83 mEq O_2_/kg (Table 1). These data indicate that the starting oil classified as “extra virgin” according to the European Regulation [12] was only slightly affected by the prolonged storage. A free acidity value ranging from 0.26 to 0.65 were found for Calabrian Ottobratica and Grossa di Gerace cultivars, respectively [25]. Higher free acidity values were recorded in EVOO from Calabrian EVOO derived from Ottobratica Calipa, Ottobratica Cannavà, Tonda di Filogaso, Ciciarello, Dolce di Rossano and Roggianella cultivars [26,27]. No significant changes were recorded in dressing free acidity since value ranging from 0.45 to 0.63% after 180 days of storage. On the contrary, according to Abedinzadeh et al. [10], who monitored the shelf-life of a dressing formulated with olive oil, apple vinegar and xanthan gum, a slight increase of the peroxide value was observed. This indicates a greater primary oxidation according to Caponio et al. [28].

Figure 2 reported the CIELab color parameters (L*, C* and H*) for both EVOO and apple vinegar-dressing monitored during storage. An increase of lightness parameter was evidenced in both samples in storage time-dependent manner. After 180 days of storage, values increased by 17.54% and 8.62% for EVOO and dressing, respectively. A similar situation was also observed in the chroma C* parameter, where a reduction by −26.42% and –18.92% for EVVO and dressing, respectively, was recorded after 180 days of storage. No significant differences were evidenced for hue angle.

According to Sikorska et al. [29] and Ceballos et al. [30], changes in EVOO color become noticeable already after the first month of storage and it is probably related to the fast photodegradation of the minor EVOO constituents with reference to pigments. In fact, positive correlations were found for all phytochemical and C* and for TCC and chlorophyll content in both EVOO and dressing, with *r* values ~1.

### 3.3. UHPLC Analysis of Agristigna EVOO and Evolution of Phytochemicals in Dressing during Storage

UHPLC analysis provided identification of individual phenols of Agristigna oil (Appendix A), but only the principal compounds were quantified (Table 2). The analyzed EVOO was characterized by a high amount of pinoresinol (49.26 mg/kg), oleuropein (25.59 mg/kg) and apigenin (16.85 mg/kg).

Previously, Brenes et al. [31] reported that the amount of pinoresinol, could be related to the hydrolysis of compounds, such as lignan linked to secoiridoid glucoside. Although EVOOs produced in the south Italy are characterized by a high content of hydroxytyrosol [32], the amount of hydroxytyrosol detected in Agristigna oil was of 2.63 mg/kg.

### 3.4. Evolution of TPC, TFC, TCC, Chlorophyll and α-Tocopherol in Dressing during Storage

Several bioactive phytochemicals were identified in Agristigna EVOO and derived products. In this study, we decided to monitor TPC, TFC, TCC, chlorophyll and α-tocopherol content, since these compounds are responsible for resistance to the oxidation, a natural decay process of EVOO quality. Moreover, both TPC and TFC are related to the pungent perception that is considered a positive EVOO sensorial attribute. The percentage of phytochemical loss as consequence of storage was calculated by considering 100% of TPC, TFC, TCC, chlorophyll and α-tocopherol content at day 0. Results are reported in Table 3.

For EVOO a significant reduction in the TPC was observed in all samples compared to the initial value. A reduction of −31.42 and −41.79% was found in dressing after 45 and 180 days of storage, respectively. A similar situation, although not as marked, was highlighted in dressing TFC content with a reduction of −14.72% after 180 days of storage. The increase of days of storage determines, like TPC, a more drastic reduction of the bioactive component referred to the TCC (−23.44% and −31.25% for 45 and 180 days of storage, respectively, compared to the value of TCC at day 0). As evidenced in Table 3, the loss of chlorophyll and α-tocopherol content is also shelf-life-dependent and reached values of −23.55 and −20.86%, respectively, after six months. α-Tocopherol is the main fat-soluble antioxidant present in EVOO and its initial content in Agristigna EVOO was 96.21 mg/kg. According to Caipo et al. [33], sample storage in the dark allows to preserve α-tocopherol by limiting its loss. Agristigna EVOO’s TPC was 390.5 mg/kg. This value is in line with those reported for different monovarietal Calabrian EVOOs, such as EVOO derived by Ottobratica and Frantoio cultivars [25,26,34]. A great variability was observed regarding TCC with values ranging from 2.26 to 7.16 mg/kg for monovarietal autochthonous Ottobratica Calipa and Ciciarello EVOO, respectively. Carotenoids are considered the most important EVOO pigments after chlorophyll due they strong protective effect against EVOO oxidation induced by light exposure. This variability was also observed in chlorophyll content [26].

A time-related deterioration in the quality of the EVOO apple vinegar-derived dressing is observed in all monitored phytochemical, where the most affected classes of compounds were: TPC (−43.77%), α-tocopherol (−32.17%) and TCC (−31.88%).

The oxidative stability data show that the induction time undergoes a reduction of 25% both in the case of the EVOO and of the dressing, which, however, is already undergoing a greater qualitative deterioration at 45 days of storage with an IT value of 26.1 h (−17.8%) (Figure 3).

### 3.5. Fatty Acids Profile in Agristigna EVOO and Its Evolution in Dressing during Storage

The fatty acid profile of Agristigna EVOO is reported in Table 4. Oleic acid (72.10%) resulted the main abundant fatty acids followed by palmitic acid (11.27%), linoleic acid (7.69%) and stearic acid (3.90%). Our data on fatty acids composition are in line with those reported in literature for different monovarietal EVOO from Calabria (Italy) [13,25,26,34].

Sicari et al. [16] reported an oleic acid content ranging from 69.6 to 76.3% for Ottobratica and Nocellara del Belice monovarietal EVOOs produced during the season 2019–2020, respectively, while values from 66.1 to 74.4% were recorded for Grossa di Gerace and Sinopolese EVOOs, respectively [25]. No significant differences among Calabrian cultivars were evidenced in palmitic acid content. Linoleic acid is more present in Agristigna EVOO than in EVOOs obtained from Sinopolese and Carolea cultivars (6.11 and 6.87%, respectively).

Glasoil sample fatty acid profile was monitored during 180 days of storage. As expected, oleic acid resulted the main abundant fatty acids followed by palmitic acid and linoleic acid (Table 4). Statistically significant differences were recorded for oleic acid and linoleic acid over time (*p* < 0.01) as well as for α-linolenic acid (from 0.48% to 0.11% at 0 and 180 days of storage, respectively). A statistically significant difference was also observed for the OA/LA ratio (*p* < 0.01), representing an indicator of oil stability [27]. Moreover, an increase of about +19.72% was observed for OA/LA during storage. Analysis of ∑SFA, MUFA and PUFA revealed that PUFA underwent a reduction by −21.71% after 180 days of storage, whereas no substantial changes were recorded for SFA and MUFA during the same period of observation. A similar situation was observed for the dressing samples formulated with olive oil, apple vinegar and xanthan gum [10]. At the same time, an increased MUFA/PUFA ratio of about +25.69% was observed at the end of observational period. *Pearson’s* correlations coefficient revealed that the ∑SFA, MUFA and PUFA were positively correlated with the content of bioactive phytochemicals (*r* > 0.8).

### 3.6. Textural Analysis

All rheological parameters showed a declining behavior over time (Table 5). Direct rheological parameters showed higher decrease percentage values (0.60% after 45 days and 4.68% after 180 days for firmness, and 2.73% after 45 days and 10.40% after 180 days for cohesiveness), while consistency showed a lower decrease rate (0.97% after 45 days and 2.09% after 180 days). It is, however, worth mentioning that the cohesion index at the 45-day test showed a mere 1.65% reduction, but after 180 days, it did not follow the other parameters behavior, with a 10.40% reduction.

### 3.7. Sensory Analysis

Glasoil samples were evaluated in terms of their sensory attributes, as presented in Figure 4. The recorded scores revealed that the storage did not affect the sensory attributes of the products, other than appearance. It should be noted that the brightness attributes seemed to decrease during storage simultaneously with the increment of browning attributes. The main olfactory descriptors referred to typical oil aroma as panelists defined the taste of the dressing as typical, with a score of 6. The bitterness taste sensation was not perceived, whereas the aftertaste was slightly recognized with a score of 2. These results suggest that the lipid oxidation did not affect the sensorial perception of the product, which preserves an acceptable sensory perception during storage.

## 4. Conclusions

With developing scientific knowledge, the consumer’s lifestyle is also getting changed principally in terms of awareness of health and several chronic diseases. The request of foods containing natural constituents promoting health benefits and possibly based on cultural flavors is now a priority. Additionally, the COVID-19 crisis has had a significant effect on consumers’ choices, boosting processed foods consumption sold exploiting e-commerce channels. In this sense, the dressings market has evolved accordingly with many products on the market meeting the consumers’ need of healthy and tasty condiments.

In this study, we presented a vinaigrette-like product (i.e., Glasoil) mainly composed of a monovarietal EVOO from Agristigna cv, an autochthonous cultivar from the Calabria region (southern Italy). The dressing was composed only by EVOO, apple vinegar and lecithin. After proper formulation, the emulsion showed good stability and viscosity over 180 days of storage. During shelf-life, both EVOO and Glasoil showed a limited oxidation of fatty acids, confirmed by the reduction of secondary metabolites. The emulsion rheological parameters get worse over time. However, sensory scores were only slightly affected during storage, as confirmed by panel tests.

Future studies will be necessary to investigate if by adding different polysaccharides, such as gum arabic or propylene glycol alginate, it is possible to improve emulsion stability over 6 months storage. Furthermore, in order to meet consumer’s needs additional studies will have to be carried out to evaluate the enrichment of dressing formulations with organic or gluten-free ingredients or flavored EVOO.

## Figures and Tables

**Figure 1 foods-11-01113-f001:**
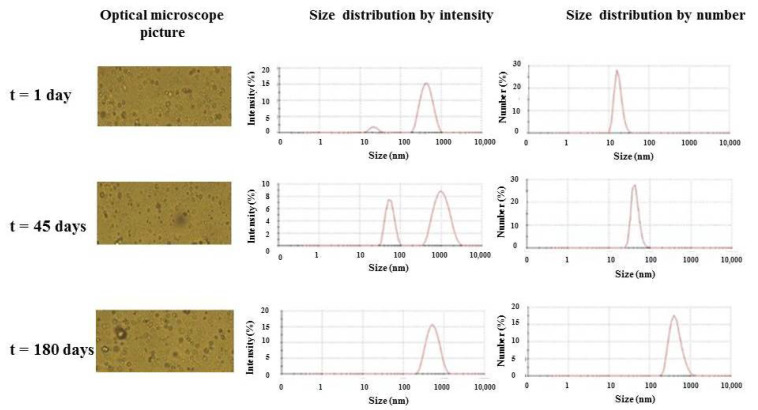
Optical microscope pictures, size distribution by intensity and number of *w*/*o* emulsion as a function of time.

**Figure 2 foods-11-01113-f002:**
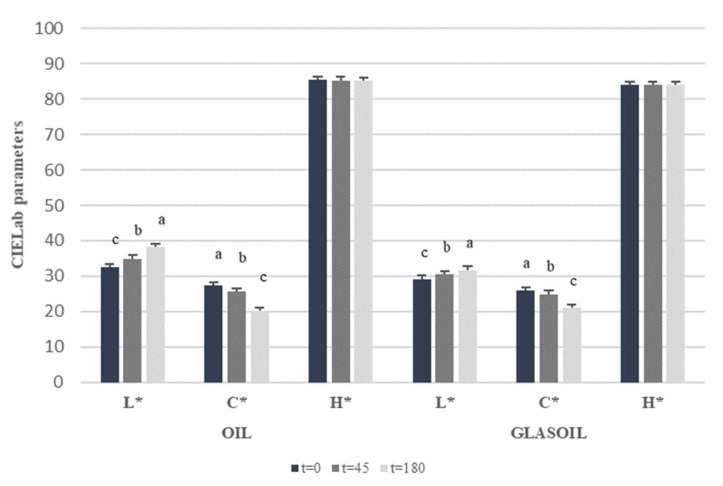
CIELab parameters of Agristigna EVOO and Glasoil Dressing evolution during 180 days of storage. Differences were evaluated by one-way analysis of variance (ANOVA) test completed with a multicomparison Tukey’s test. Means in the same parameter with different small letters differ significantly.

**Figure 3 foods-11-01113-f003:**
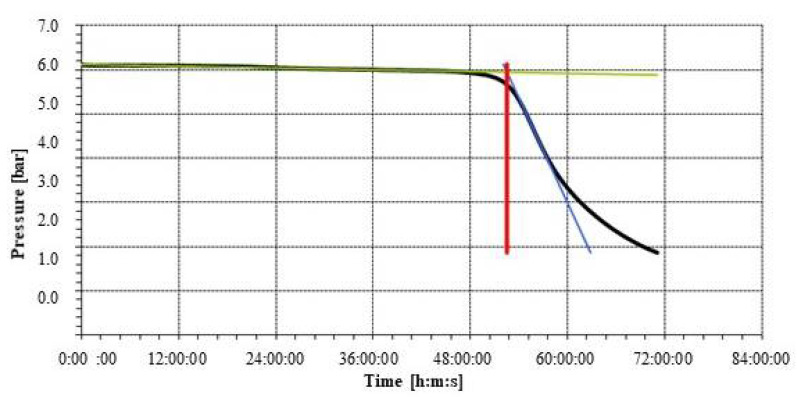
Agristigna EVOO induction time.

**Figure 4 foods-11-01113-f004:**
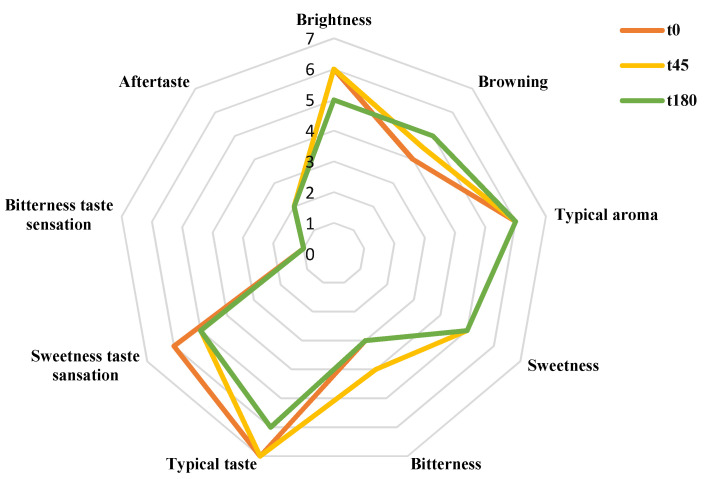
Effect of storage on specific sensory attributes of Agristigna EVOO apple vinegar-based dressing. A 10-point structured scales, where 0 is absent and 10 is extremely perceptible was used. Sensorial parameters were monitored at 0, 45 and 180 days storage.

**Table 1 foods-11-01113-t001:** Comparative quality parameters of Agristigna EVOO and Glasoil during shelf-life (180 days of storage in dark at 4 °C).

Sample	Quality Index	Days of Storage			
		0 Days	45 Days	180 Days	Sign.
Agristigna EVOO	Free acid (%)	0.31 ± 0.12 ^b^	0.33 ± 0.16 ^b^	0.52 ± 0.23 ^a^	**
	Peroxide value (mEq O_2_/kg)	12.14 ± 1.14 ^c^	14.62 ± 1.18 ^b^	17.83 ± 1.19 ^a^	**
	Induction time (h)	54.40 ± 0.98 ^a^	49.73 ± 0.78 ^b^	40.56 ± 0.87 ^c^	**
Glasoil Dressing	Free acid (%)	0.45 ± 0.16 ^b^	0.47 ± 0.18 ^b^	0.63 ± 0.24 ^a^	**
	Peroxide value (mEq O_2_/kg)	17.46 ± 1.22 ^b^	17.72 ± 1.24 ^b^	19.99 ± 1.25 ^a^	**
	Induction time (h)	34.87 ± 8.76 ^a^	28.65 ± 0.65 ^b^	26.12 ± 0.34 ^c^	**

Sign.: significant. Differences were evaluated by one-way analysis of variance (ANOVA) test completed with a multicomparison Tukey’s test. ** *p* < 0.01. Means in the same row with different small letters differ significantly.

**Table 2 foods-11-01113-t002:** Phenolic composition of monovarietal Agristigna EVOO produced in Calabria (Italy) during the 2018–2019 season.

Compound	Amount in EVOO (mg/kg)
	Agristigna
Hydroxytyrosol	2.63 ± 0.01
Tyrosol	8.58 ± 0.02
*p*-Cumaric acid	0.23 ± 0.00
Oleuropein	25.59 ± 0.51
Luteolin	3.64 ± 0.07
Pinoresinol	49.26 ± 0.28
Apigenin	16.85 ± 0.23

Data are presented as mean ± standard deviation (S.D.).

**Table 3 foods-11-01113-t003:** Comparative bioactive phytochemical content (mg/kg) of Agristigna EVOO and Glasoil dressing during shelf-life (180 days of storage in the dark at 4 °C).

Sample	Bioactives	Day’s Storage			
		0 Days	45 Days	180 Days	Sign.
EVOO	TPC	390.5 ± 4.9 ^a^	267.8 ± 4.4 ^b^	227.3 ± 4.0 ^c^	**
	TFC	19.7 ± 2.5 ^a^	19.3 ± 2.2 ^a^	16.8 ± 2.0 ^b^	**
	TCC	6.4 ± 0.5 ^a^	4.9 ± 0.3 ^b^	4.4 ± 0.3 ^b^	**
	Chlorophyll	3.4 ± 0.4 ^a^	3.2 ± 0.3 ^a^	2.6 ± 0.2 ^b^	**
	*α*-Tocopherol	96.4 ± 1.6 ^a^	82.3 ± 1.5 ^b^	76.1 ± 1.2 ^c^	**
Glasoil Dressing	TPC	482.7 ± 4.8 ^a^	289.8 ± 5.1 ^b^	271.4 ± 4.9 ^c^	**
	TFC	25.9 ± 2.8 ^a^	22.7 ± 2.5 ^b^	19.7 ± 2.4 ^c^	**
	TCC	6.9 ± 0.7 ^a^	6.5 ± 0.5 ^a^	4.7 ± 0.6 ^b^	**
	Chlorophyll	3.23 ± 0.6 ^a^	3.2 ± 0.3 ^a^	2.7 ± 0.2 ^b^	**
	α-Tocopherol	121.9 ± 3.1 ^a^	101.4 ± 2.8 ^b^	82.7 ± 1.3 ^c^	**

TPC: Total phenols content; TFC: Total flavonoids content; TCC: total carotenoids content. Sign.: significant. Differences were evaluated by one-way analysis of variance (ANOVA) test completed with a multicomparison Tukey’s test. ** *p* < 0.01. Means in the same row with different small letters differ significantly.

**Table 4 foods-11-01113-t004:** Fatty acid profile of Agristigna EVOO and Glasoil dressing during storage.

Fatty Acid	EVOO	GLASOIL DRESSING			
		Days			
		0	45	180	Sign.
Myristic acid	0.01 ± 0.003 ^a^	0.02 ± 0.005 ^a^	ND	ND	**
Palmitic acid	11.27 ± 0.65 ^b^	12.15 ± 0.76 ^a^	12.10 ± 0.62 ^a^	12.08 ± 0.56 ^a^	**
Palmitoleic acid	0.48 ± 0.04 ^b^	0.56 ± 0.07 ^a^	0.46 ± 0.05 ^b^	0.34 ± 0.02 ^c^	**
Margaric acid	0.06 ± 0.04 ^a^	0.06 ± 0.07 ^a^	ND	ND	**
Margaroleic acid	0.72 ± 0.05 ^a^	0.78 ± 0.03 ^a^	0.71 ± 0.09 ^a^	0.54 ± 0.08 ^c^	**
Stearic acid	3.90 ± 0.22 ^a^	3.93 ± 0.24 ^a^	3.65 ± 0.19 ^b^	3.58 ± 0.14 ^c^	**
Oleic acid	72.10 ± 0.78 ^a^	72.16 ± 0.83 ^a^	71.81 ± 0.85 ^b^	71.63 ± 0.78 ^b^	**
Linoleic acid	7.69 ± 0.22 ^b^	7.73 ± 0.29 ^a^	7.68 ± 0.27 ^b^	6.32 ± 0.19 ^a^	**
Arachidic acid	0.63 ± 0.05 ^a^	0.64 ± 0.08 ^a^	0.63 ± 0.05 ^a^	0.57 ± 0.04 ^b^	**
α-Linolenic acid	0.43 ± 0.02 ^a^	0.48 ± 0.06 ^a^	0.38 ± 0.07 ^b^	0.11 ± 0.07 ^c^	**
Gadoleic acid	0.38 ± 0.02 ^a^	0.37 ± 0.06 ^a^	ND	ND	**
Behenic acid	0.10 ± 0.03 ^a^	0.11 ± 0.02 ^a^	ND	ND	**
Lignoceric acid	0.02 ± 0.003 ^a^	0.02 ± 0.002 ^a^	ND	ND	**
OA/LA	9.37	9.33	9.35	11.17	
∑SFA	15.99	16.29	15.75	15.66	
∑MUFA	73.70	73.09	72.27	71.97	
∑PUFA	8.12	9.63	9.40	7.54	
MUFA/PUFA	9.07	7.59	7.69	9.54	

Data are expressed as the mean ± standard deviation (SD) (*n* = 3). ND: not detected. Sign.: significant. Differences were evaluated by one-way analysis of variance (ANOVA) completed with a multicomparison Tukey’s test. ** *p* < 0.01. Means in the same row with different small letters differ significantly.

**Table 5 foods-11-01113-t005:** Textural parameters of Agristigna EVOO-apple cider-derived dressing.

Parameters	Days of Storage			Sign.
	0 Days	45 Days	180 Days	
Firmness (g)	16.65 ± 1.15 ^a^	16.55 ± 1.01 ^a^	15.87 ± 0.95 ^b^	**
Consistency (g∙s)	221.51 ± 12.45 ^a^	219.36 ± 12.08 ^b^	216.88 ± 11.15 ^c^	**
Cohesiveness (g)	−7.69 ± 0.12 ^b^	−7.48 ± 0.08 ^c^	−6.89 ± 0.05 ^a^	**
Cohesion index (g∙s)	18.19 ± 1.25 ^a^	17.89 ± 1.13 ^b^	16.87 ± 1.10 ^c^	**

Data are expressed as the mean ± standard deviation (SD) (*n* = 3). Sign: significant. Differences were evaluated by one-way analysis of variance (ANOVA) completed with a multicomparison Tukey’s test. ** *p* < 0.01. Means in the same row with different small letters differ significantly.

## Data Availability

All data and materials are available on request to the corresponding author.

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
