# Peer review of "Evaluation of Selected Quality Parameters of “Agristigna” Monovarietal Extra Virgin Olive Oil and Its Apple Vinegar-Based Dressing during Storage"

_foods, 2022, doi:10.3390/foods11081113_

Round 1
Reviewer 1 Report
The manuscript entitled “Evaluation of selected quality parameters of “Agristigna” monovarietal extra virgin olive oil and its apple-vinegar based 3 dressing during storage” investigated the stability of 3 dressings during storage. Generally, this work seems worthy of investigation. Many effect factors were detected and detailed detection procedures were also provided. Only the sensory evaluation section should be revised to be clear. Therefore, I recommend minor revision.
Here are the specific suggestions
- Introduction: what’ve previous works (main analytical methods and main conclusions or findings) done in characterization in the stability of salad dressings? This section lacked introduction. Besides, the limitations of the current studies should also be discussed in the introduction section. More references should be cited.
- Punctuation is lacking at the end of Line 45.
- Line 52 “https”?
- Line 112: ppm should be revised to mg/kg; please check the whole manuscript.
- Line 159-166: the sensory evaluation should be a separate paragraph and the sensory evaluation methods should be detailed described, including the panelist training, descriptor description and corresponding reference compounds. Please cite the aroma, taste and texture characteristics of foods from Pu et al., Liang et al. and also other related references. (Food Chemistry, 2022, 383, 132455.; Food Chemistry, 2021, 339, 128078.; Food Chemistry, 2020, 318, 126520.; LWT, 2021, 138, 110641.)
- The inconsistent number of valid digits in Table 1, 2, and 3 should be revised.
- Figure 3: what’s the typical aroma? The line of each color should be revised to different samples, whereas the sensory attributes should be listed at the triangle of each figure. Please revise the Figure 3.
- Line 184, Tara gum, “T” should be lowercase font.
- The title of chapter 3.2. 3.3, and 3.4 are too complex. Please simplify them.
- Line 225, ” meq O2/kg ”?
- Line 375 and 377, “p” should be italicized.
- The character in Fig.3 should be black (Gray looks unclear). Also, the character in all figures should keep unified.
Author Response
Reviewer 1
The manuscript entitled “Evaluation of selected quality parameters of “Agristigna” monovarietal extra virgin olive oil and its apple-vinegar based 3 dressing during storage” investigated the stability of dressings during storage. Generally, this work seems worthy of investigation. Many effect factors were detected and detailed detection procedures were also provided. Only the sensory evaluation section should be revised to be clear. Therefore, I recommend minor revision.
Here are the specific suggestions
Q1: Introduction: what’ve previous works (main analytical methods and main conclusions or findings) done in characterization in the stability of salad dressings? This section lacked introduction. Besides, the limitations of the current studies should also be discussed in the introduction section. More references should be cited.
A1: The scientific literature based on the study of the shelf-life of dressings formulated with different oils appears somewhat limited. However additional references were opportunely cited to improve Introduction section see lines 62-85.
Q2: Punctuation is lacking at the end of Line 45.
A2: We have checked and corrected.
Q3: Line 52 “https”?
A3: We have checked and corrected typos with reference 5.
Q4: Line 112: ppm should be revised to mg/kg; please check the whole manuscript.
A4: We have checked and replaced “ppm” with “mg/kg” in all text.
Q5: Line 159-166: the sensory evaluation should be a separate paragraph and the sensory evaluation methods should be detailed described, including the panelist training, descriptor description and corresponding reference compounds. Please cite the aroma, taste and texture characteristics of foods from Pu et al., Liang et al. and, also other related references. (Food Chemistry, 2022, 383, 132455.; Food Chemistry, 2021, 339, 128078.; Food Chemistry, 2020, 318, 126520.; LWT, 2021, 138, 110641.)
A5: We have separated the paragraph regarding the Sensory analysis. Moreover, all requested details are inserted following Reviewer’s suggestion.
Q6: The inconsistent number of valid digits in Table 1, 2, and 3 should be revised.
A6: We have checked and corrected number in all tables. Average diameters and scattering intensities were modify in the text of the revised manuscript according to the instrument (Zetasizer Nano ZS instrument (Malvern 89 Panalytical Ltd, Malvern, United Kingdom)) resolution.
Q7: Figure 3: what’s the typical aroma? The line of each color should be revised to different samples, whereas the sensory attributes should be listed at the triangle of each figure. Please revise the Figure 3.
A7: We have checked and revised all section regarding sensory analysis. The attribute “typical aroma” was used to test the sensory impact of adding apple vinegar to oil in order to define the difference with “conventional” aromatic perceptions of EVOO.
As suggested Figure 4 was revised.
Q8: Line 184, Tara gum, “T” should be lowercase font.
A8: We have checked and corrected.
Q9: The title of chapter 3.2. 3.3, and 3.4 are too complex. Please simplify them.
A9. We have simplified the paragraph titles.
Q10. Line 225, ” meq O2/kg ”?
A10: We have checked and corrected.
Q11: Line 375 and 377, “p” should be italicized.
A11: We have checked and corrected. Now “p” is in italics.
Q12: The character in Fig. 4 should be black (Gray looks unclear). Also, the character in all figures should keep unified.
A12: We have replaced Figure 4 with a new one and we have standardized the font in all the figures in the manuscript. As suggested the characters were changed (black color) and unified in all figures.
Moreover, we have done corrections of typos in all manuscript.

Reviewer 2 Report
The manuscript is professionally written and organized. I recommended that it was published after minor changes:
Figure 3 did not provide any significant information; it was extremely hard to take any information form it. Thus, take it off or shown the results
Table 2, there was no sense in use Tukey test, you just have one group variable.
Add standard deviation in Table 1.
Material and methods, Excel do not do the Tukey test. Please, explain how the Tukey test was carried out using Excel.
Author Response
Reviewer 2
The manuscript is professionally written and organized. I recommended that it was published after minor changes:
Q1: Figure 3 did not provide any significant information; it was extremely hard to take any information form it. Thus, take it off or shown the results.
A1: We have replaced Figure 3 with a new one to make the results obtained clearer.
Q2: Table 2, there was no sense in use Tukey test, you just have one group variable.
A2: Following Reviewer’s suggestion we have removed Tukey test from Table 2.
Q3: Add standard deviation in Table 1.
A3: We have added SD to Table 1.
Q4: Material and methods, Excel do not do the Tukey test. Please, explain how the Tukey test was carried out using Excel.
A4: Tukey test was performed using SPSS software for Windows (SPSS Inc., El-gin, IL, U.S.A.) 22.0 Version. Microsoft Excel 2010 software was used to calculate linear regression, assessment of repeatability, calculation of average, relative standard deviation, and Pearson’s correlation coefficient (r).
Moreover, we have done corrections of typos in all manuscript.

Reviewer 3 Report
Overall, the introduction, material and methods, results, and discussion, as well as conclusions are properly presented, confirming the high scientific expertise of the research team. The main objective of this study was the evaluation of some quality and shelf-life features of an Agristigna EVOO-apple-vinegar based dressing over 180 days storage at 4°C. The research work topic is important and worth of investigation and approving, however there are many shortcomings which must be rectified. In general, important information is presented.
- Please do not use abbreviation in the abstract
2. This works has a variety of data which are not apparent by just reading the abstract. There appears to be some information, which can add to knowledge in this growing field.
- L46- L52, L73-L82: please give a reference
- Introduction need more investigation, in fact the introduction did not provide sufficient background and need to include more references
- L210-L217 – 378-L388 and L415-L417, please give more explication
- In the discussion, authors would have benefited from a better understanding of the existing literature.
- In some sentences, English appears not to be adequate.
- To use a space between the number and the unit, as 20 °C; and not to use a space between number and percentage, like 10%, for example.
- Conclusion: please include limitations and future research areas!
- References must be revised.
Author Response
Reviewer 3
Overall, the introduction, material and methods, results, and discussion, as well as conclusions are properly presented, confirming the high scientific expertise of the research team. The main objective of this study was the evaluation of some quality and shelf-life features of an Agristigna EVOO-apple-vinegar based dressing over 180 days storage at 4°C. The research work topic is important and worth of investigation and approving, however there are many shortcomings which must be rectified. In general, important information is presented.
Q1: Please do not use abbreviation in the abstract. This works has a variety of data which are not apparent by just reading the abstract. There appears to be some information, which can add to knowledge in this growing field.
A1: We have removed abbreviation from the abstract section.
Q1: L46- L52, L73-L82: please give a reference
A1: We have added reference 5 to lines 46-52; it is not possible to insert any reference in lines 73-82 because the preparation of emulsion in this way is published here for the first time.
Q3: Introduction need more investigation; in fact in the introduction did not provide sufficient background and need to include more references.
A3: We have added data on state of art in lines 62-85.
Q4: L210-L217 – 378-L388 and L415-L417, please give more explication.
A4: We have improved lines 210-217, 378-388 and 415-417.
Q5: In the discussion, authors would have benefited from a better understanding of the existing literature.
A5. We have added some comments from previously published literature.
Q6: In some sentences, English appears not to be adequate.
A6: We have checked manuscript for English language.
Q7: To use a space between the number and the unit, as 20 °C; and not to use a space between number and percentage, like 10%, for example.
A7: We have checked and corrected.
Q8: Conclusion: please include limitations and future research areas!
A8: We have improved Conclusion section.
Q9: References must be revised.
A9: We have checked and corrected References following MDPI style.
